# Exploring the Oral Microbiome in Rheumatic Diseases, State of Art and Future Prospective in Personalized Medicine with an AI Approach

**DOI:** 10.3390/jpm11070625

**Published:** 2021-06-30

**Authors:** Silvia Bellando-Randone, Edda Russo, Vincenzo Venerito, Marco Matucci-Cerinic, Florenzo Iannone, Sabina Tangaro, Amedeo Amedei

**Affiliations:** 1Department of Clinical and Experimental Medicine, University of Florence, Largo Brambilla 3, 50134 Florence, Italy; silvia.bellandorandone@unifi.it (S.B.-R.); edda.russo@unifi.it (E.R.); marco.matuccicerinic@unifi.it (M.M.-C.); 2Rheumatology Unit, Department of Emergency and Organ Transplantations, University of Bari “Aldo Moro”, 70121 Bari, Italy; vincenzo.venerito@uniba.it (V.V.); florenzo.iannone@uniba.it (F.I.); 3Unit of Immunology, Rheumatology, Allergy and Rare Diseases (UnIRAR), IRCCS San Raffaele Hospital, 20132 Milan, Italy; 4Dipartimento Interateneo di Fisica “M. Merlin”, Istituto Nazionale di Fisica Nucleare, Sezione di Bari, 70121 Bari, Italy; sabina.tangaro@uniba.it

**Keywords:** oral microbiota, microbiome, rheumatology diseases, biomarkers, artificial intelligence, machine learning, rheumatoid arthritis, Sjogren’s syndrome, systemic lupus erythematosus

## Abstract

The oral microbiome is receiving growing interest from the scientific community, as the mouth is the gateway for numerous potential etiopathogenetic factors in different diseases. In addition, the progression of niches from the mouth to the gut, defined as “oral–gut microbiome axis”, affects several pathologies, as rheumatic diseases. Notably, rheumatic disorders (RDs) are conditions causing chronic, often intermittent pain affecting the joints or connective tissue. In this review, we examine evidence which supports a role for the oral microbiome in the etiology and progression of various RDs, including rheumatoid arthritis (RA), Sjogren’s syndrome (SS), and systemic lupus erythematosus (SLE). In addition, we address the most recent studies endorsing the oral microbiome as promising diagnostic biomarkers for RDs. Lastly, we introduce the concepts of artificial intelligence (AI), in particular, machine learning (ML) and their general application for understanding the link between oral microbiota and rheumatic diseases, speculating the application of a possible AI approach-based that can be applied to personalized medicine in the future.

## 1. Introduction

The human body is a symbiotic ecosystem containing trillions of microorganisms classified into the domains of the *Eukarya* (and their viruses), *Bacteria*, *Archaea*. These communities have been named as microbiota, while the term microbiome describes either the collective genome of the microorganisms that reside in an environmental niche. The microbiota is on all surfaces of our body exposed to the external environment, from the respiratory to the gastrointestinal and urogenital tract.

A balanced composition of the microbiome is of paramount significance, influencing healthy and pathological states through many biological activities as regulation of metabolic processes, energy extraction, defense from pathogenic microorganisms, production of vitamins, and modulation of the immune system [1,2,3,4]. A healthy microbiome is described by high diversity, while the loss of variety may lead to “dysbiosis”, a crucial condition of disequilibrium between pathogenic and commensal bacteria. Human microbiota has been studied in several tissue sites as skin, oral, subgingival, nasal, lung, and vessels.

Currently, the oral microbiome is receiving growing interest from the scientific community, as the mouth is the gateway for different potential etiopathogenetic factors in numerous diseases. Thus, also the commensal oral microbiome has an important role in the maintenance of oral and systemic health. Due to the abundance of nutrients, the human oral cavity represents an ideal habitat for the most varied and distinctive communities of microorganism in the human body, however, to date, these communities remain relatively understudied as compared to the gut ones [3,5]. The ecological oral niche consists of five distinct areas: saliva, teeth, tongue, gingival sulcus and periodontal pocket, and the remaining oral mucosa. Concerning the microbiota composition of the oral niches, it has been observed that the mouth bacterial microflora is organized as a highly interconnected chain of microbiomes across the human body, creating a kind of micro-biosphere, composed of small multiple ecological niches with different groups of microorganisms [6]. The progression of niches from the mouth to the intestine is defined as “oral–gut microbiome axis”, rather that the oral microbiome alone, it is the complexity of the interrelationships between the different microbiomes through this axis that affects the healthy status. However, transfer of oral bacteria to the gut is therefore common. Members of the oral and oropharyngeal microbiota reach the gut through swallowed saliva, nutrients, and drinks.

Multiple reports on the composition of the microflora within the oral niches show higher levels of the genus *Corynebacterium* in gingival plaques [7] and higher levels of the phylum *Firmicutes* in buccal mucosa and saliva [8]. In addition, fungi have also been detected as members of the healthy oral microbiota (belonging to the oral Mycobiota) [9].

It has been observed that several factors are the main causes for oral microflora dysbiosis, including poor oral hygiene, dietary habits, gingival inflammation, dysfunction of the salivary glands, smoking, genetic difference [10,11]. In addition, changes in the salivary concentration of nutrients, oxygen, and pH can induce the selection of different microorganisms involved in the formation of their own niche via biofilm development or nutrient metabolism [12]. However, a dysbiotic oral microbiome is involved in a number of oral infectious diseases, such as periodontitis, dental caries, alveolar bone loss, endodontic infection, and tonsillitis [13], and finally seem to be involved in pathogenesis of more systemic diseases [14,15,16]. It has been speculated that the dysbiosis in the oral cavity is guided by “keystone pathogens”, which can regulate community microbiome variations. Due to the high vascularity of the mouth and the manipulation of a host response, the dysbiotic oral microflora can influence the activities at other body districts. As previously cited, this condition suggests potential implications of the oral microbiome not only in oral diseases but also in a number of systemic pathologies [17,18,19] and notably, several cancer types [20]. Recently, many scientists have focused on the potential role of the commensal bacteria in the pathogenesis of systemic autoimmune diseases as rheumatic diseases (RDs).

## 2. The Oral Microbiome in Rheumatic Diseases

The etiopathogenesis of systemic autoimmune diseases is still almost unknown, but a complex interplay of environmental and genetic factors associated at stochastic events lead to loss of immunological tolerance and autoimmunity [21]. Numerous advances in understanding RDs pathogenesis have provided a mechanistic framework for studying host–microbiota interactions and candidate pathobionts. The gut microbiome is the most deeply evaluated also in RDs but several studies of the human skin, oral, subgingival, nasal, lung, and vascular microbiota suggest that dysbiosis is a common feature across rheumatic diseases, in particular, rheumatoid arthritis (RA), connective tissue diseases, and primary vasculitides and it might impact their symptoms and disease course [22] (Figure 1).

The introduction of high-throughput methods in microbiome analysis like sequencing of 16S rRNA, NGS (Next Generation Sequencing), and metabolomics gave the possibilities for large cohort studies which showed that microbiota may have a protective, neutral, or provocative role in the context of autoimmunity as Yurkovetskiy et al. pointed out [23].

For example, commensals may have direct effects as the well established molecular mimicry that may be a potential ligand to autoimmunity through the existence of non-selective T cell receptors with cross-reactivity to self-antigens [23,24,25].

Moreover, in recent years, saliva analysis has also played a central role in the definitions of biomarkers for the diagnosis, prognosis, and treatment of oral and systemic disease. Recent advances in salivary biomarker diagnostics have broadened the discovery of microbial pathogens associated with systemic and oral diseases [26]. While host biomarkers are subjected to individual biological variation, oral microbiome is relatively conserved among unrelated individuals. Indeed, the analysis of the oral microbial changes could be considered in the future as a screening biomarker also for rheumatic diseases, leading to a personalized medicine approach. In this review, we will focus on the oral–gut microbiota axis and we will evaluate the most recent studies endorsing it as a promising diagnostic biomarker for RDs; in addition, we will explore new perspectives in microbiome research trough a machine learning approach.

## 3. Oral Microbiome and Rheumatoid Arthritis

Rheumatoid arthritis (RA) is a systemic chronic autoimmune inflammatory disease characterized by destruction of bone in multiple joints and auto-antibody production such as rheumatoid factor and in particular, anticitrullinated protein antibodies (ACPAs), the most powerful and earlier diagnostic biomarkers [27,28,29]. Periodontal disease (PD) and tobacco use seem to be related to ACPA production, RA etiopathogenesis still remains unclear. The idea that oral microbiota is involved in inducing or driving of RA progression is supported by the high frequency of periodontal inflammatory disorders in RA patients and its association with anti-CCP antibody levels [30] (Figure 2).

In addition, animal models have reported that RA alters, qualitatively and quantitatively, the oral microbiome, highlighting the bidirectional link between host and microbiota [31]. The discovery that post-translational protein citrullination primarily determines autoantibody reactivity in RA has clarified microbial contributions to its pathogenesis [32].

Periodontitis is characterized by a chronic inflammation caused by oral bacteria and leucocyte infiltration with progressive destruction of the alveolar bone and it seems to share the same pathogenetic mechanisms with RA: accumulation of leucocyte infiltration, release of inflammatory cytokines, and mediators such as prostaglandin E2 (PGE2), tumour necrosis factor (TNF)-𝛼, and several other citokines, such as interleukin (IL)-1b, IL-6, IL-17, IL-33, IL-12, IL-18r [33]. In a recent study, salivary cytokine analysis showed that IL-17 was markedly increased in RA patients with periodontitis, but not periodontitis without RA [34].

*P. gingivalis* is considered the keystone pathogen in periodontitis and the most extensively studied in RA and it was found to deregulate local immune responses and to promote dysbiosis. The clinical association of *P. gingivalis* and RA has been investigated by many studies, most commonly by molecular detection of bacterial DNA from plaque or gingival crevicular fluid, or by bacterial culture, by measuring serum antibody reactivity and its role in the development of experimental arthritis, has been explored in animal models [35,36]. *P. gingivalis* was first proposed as a crucial agent in RA onset, expressing a bacterial protein arginine deiminase (PPAD) that can citrullinate free L-arginine and C-terminal arginine residues in cleaved peptides [37]. C-terminal citrullination of both bacterial and host protein has been hypothesized to break immunological tolerance and initiate the ACPA response in RA [38]. Sato et al. [39] also showed that *P. gingivalis* exacerbate arthritis by modulating the gut microbiota and increasing the proportion of Th17 (T helper 17) cells in mesenteric lymph nodes. Another proof of the link between RA and oral inflammation is the evidence that the treatment of periodontal disease may improve RA symptoms [40].

Furthermore, microbiome of periodontally healthy individuals with and without RA has been evaluated. The subgingival microbiota differed significantly between RA and healthy individuals. In the absence of periodontitis, *Porphyromonas gingivalis* and *A. actinomycetemcomitans* did not differ significantly between groups [41]. In contrast, *Cryptobacterium curtum* was increased in RA patients [35]. *Aggregatibacter actinomycetemcomitans*, a periodontal bacterium, was recently proposed to connect periodontitis to RA for its ability to induce citrullinated autoantigens through the bacterial pore-forming toxin leukotoxin A (LtxA), which is its primary virulence factor [42]. LtxA induces the citrullination of a wide range of RA autoantigens which are subsequently released by the dying neutrophil in a process that is reminiscent of NETosis but is biologically distinct. Exposure to *A. actinomycetemcomitans* by anti-LtxA antibody reactivity was observed in a large RA subset [42], a finding that has been replicated in an independent Dutch cohort [43,44]. HLA-DRB1 shared epitope risk alleles associated with ACPAs only in RA patients with evidence of *A. actinomycetemcomitans* exposure, suggesting that LtxA-induced protein citrullination may play a role in ACPA production in genetically susceptible subjects [42].

Several other bacteria have been implicated in RA pathogenesis. Recently, Brusca et al. [45] found that there were more organisms besides *P. gingivalis* which cause periodontal disease (i.e., *Anaerglobus geminatus* and *Prevotella/Leptotrichia*) and were linked to the ACPA presence.

In the oral flora, bacteria such as *P. intermedia**/Tannerella forsythia* were found and high titers of antibodies against these microorganisms have been detected in the serum and synovial fluids of RA patients [46]. Otherwise, IgG antibodies to *P. intermedia* and *C. ochracea* were found to be associated with a lower RF prevalence [47,48]. Oral microbiota seems to have a crucial role not only in RA pathogenesis but also in exacerbation of joint involvement and in treatment response. In fact, it has been shown that dysbiosis may exacerbate arthritis through the oral inoculation of *P. gingvalis* using animal models, even if the mechanism underlying the exacerbation is not entirely elucidated [49]. In a collagen-type II experimental arthritis in mice, periodontitis caused by *P. gingivalis* and *Provotellanigrescens* exacerbated arthritis through TLR2-dependent antigen-specific Th17 immune response [49].

Regarding oral microbiota in early RA patients compared to those with advanced phases, Wolff et al. showed that patients with early RA showed a high PD prevalence at disease onset [50] and that microbiota of their subgingival biofilm was similar to that of patients with chronic RA. Scher JU et al. reported that the subgingival microbiota in patients with new-onset RA was distinct from healthy controls [51].

*Prevotella* and *Leptotrichia* were found in subgingival microbiota from new-onset RA patients but not from healthy controls and this was unrelated to periodontal disease. The abundance of *P. gingivalis* was directly associated with periodontitis severity but it did not correlate with ACPAs, while *Anaeroglobus geminatus* correlated with ACPAs and the rheumatoid factor [52]. Zhang et al., in a large Chinese study, demonstrated that changes in the dental, saliva, or gut microbiome identified RA patients from healthy controls. In particular, treatment-naive RA patients had an increased abundance of *Lactobacillus salivarius* in the microbiota of all three sites, in particular those with high disease activity [53], while *Haemophilus* species were depleted in RA patients and negatively correlated with serum levels of autoantibodies.

Mikules et al. showed that dysbiosis was partially resolved following treatment with disease modifying anti-rheumatic drugs [54], further stressing the bidirectional crosstalk between the microbiota and their hosts. Interestingly, Ceccarelli et al. demonstrated a significant association between the percentage of *P. gingivalis* on the total tongue biofilm and RA disease activity (DAS28) [55].

Furthermore, studies have shown that RA patients have altered gut and mouth microbiomes that are partly normalized after disease modifying antirheumatic drugs (DMARDs) and could predict response to treatment [56]. Zhang, X. et al. showed that patients that responded well to treatment were characterized by a greater number of virulence factors before treatment and also by the reduction in *Holdemania filiformis* and *Bacteroides* sp. after treatment. They also reported that the effect on the gut microbiome was moderate compared to the oral microbiome [53].

## 4. Link between Oral Microbiome and Sjogren’s Syndrome

Sjogren’s syndrome (SS) is an autoimmune disease characterized by autoantibody production (frequently directed against ribonucleoproteins TRIM21/Ro52/SS-A, Ro60, and La/SS-B), destruction of exocrine glands (in particular, salivary and lacrimal) by lymphocytes, and of extraglandular epithelial tissues [57]. SS can co-exist or pre-exist with other RD and then, it is known as secondary Sjogren’s syndrome. Exocrine glands infiltration causes a reduction of production of saliva and tear film, leading to an alteration of mucosal barrier function and favoring dysbiosis and colonization with pathobionts [58]. From a clinical point of view, the most frequent symptoms are dryness of mouth and eyes while extraglandular manifestations include arthritis, peripheral neuropathy, cutaneous vasculitis, respiratory dysfunction, and tubulointerstitial nephritis [59]. The condition of dry mouth imposes a very significant burden on many oral functions, such as speaking and eating, reducing greatly the life quality of patients but also making patients at a considerable risk for severe oral and dental diseases. However, SS is considered benign, even patients have an increased risk of developing lymphoma. Genetic predisposition and environmental factors are the most important etiopathogenetic SS factors [60], but, as recently suggested, also the dysbiosis may play a significant role in SS pathogenesis. Oral and gut microbiome in SS have been explored to evaluate a possible interplay between human microbiome and SS clinical manifestations and disease severity. In fact, SS alters the saliva composition, which in turn induces alterations in the oral microbiome, certain organisms e.g., *Capnocytophaga*, *Dialister*, *Fusobacterium*, *Helicobacter*, *Streptococcus*, and *Veilonella* were found in abundance in SS, while *Porphyromona sgingivalis* and *Actinobacillus actinomycetemcomitans* were not detected in any SS patient [61]. On the other hand, it has been hypothesized that dysbiosis and bacterial translocation propagate local and systemic inflammation, modulating SS severity (Figure 3).

In recent years, there have been five studies focused on SS microbiome and exploring different oral niches: one each from saliva [62], tongue [63], and oral washings [64] and two from buccal mucosa [65]. Interesting results have been reported on oral microbiota composition in primary SS (pSS). Van der Meulen et al. [14] examined the bacterial composition by 16S rRNA sequencing of the oral microbiome in 37 pSS patients, 86 non-SS sicca patients, and 24 healthy controls. They found that buccal mucosa microbiome of pSS and non-SS sicca patients differed from healthy controls, with a higher *Firmicutes/Proteobacteria* ratio observed in both SS and non-SS sicca patients. These data highlight how oral dryness may favor dysbiosis. In a similar study, evaluating chewing-stimulated whole saliva *Porphyromonas endodontalis*, *Prevotella intermedia*, *Fusobacterium nucleatum vincentii*, *Streptococcus intermedius*, *Tannerella* spp., and *Treponema* spp. were only detected in the context of oral dryness (SS and non-SS sicca patients), while *Porphyromonas pasteri* showed increased abundance in healthy controls [66]. De Pavia et al. found that patients with primary SS had increased levels of *Lactobacillus* spp., *S. mutans*, and *Candida albicans* within their supragingival plaque samples. Samples from the oral mucosa and tongue also showed increased prevalence of *Staphylococcus* aureus and *Candida albicans*. The greatest shift in microbial differences was observed in patients who had a saliva reduction [63]. Furthermore, increasing evidence suggests a role for cross-reactivity of commensal oral and gut bacteria with SSA/Ro60 in the SS etiopathogenesis [67]. The key role of activated B cells has been established in the pathogenesis of Sjogren’s syndrome. Their activation leads to the production of autoantibodies and hypergammaglobulinemia seen in some SS patients [68]. Moreover, it has been demonstrated that the von Willebrand factor type A, a microbial protein shared by different commensal oral/gut bacteria, could activate Ro60 reactive T cells, promoting autoantibody responses against Ro60. In fact, *Corynebacterium amycolatum* has been shown to colonize the lacrimal duct, making C. amycolatum Ro60 a candidate ortholog for the development of anti-Ro60 antibodies in SS [69].

It is possible that a dysregulated immune response against the normal microbiome can be one of the potential pathways starting the autoimmune responses in SS. Considering that oral infections are a common problem in SS patients, this pathway might also be involved in amplification of autoimmune responses in this disease [70]. However, whether oral dysbiosis is merely a consequence of oral dryness and decreased antimicrobial properties of saliva or an active driver of systemic and target tissue inflammation in patients with SS is still unknown. The microbiome of the buccal mucosa is not specific enough for pSS and therefore not useful for characterizing SS patients in clinical practice. It is unlikely that one specific bacterial taxon in the buccal mucosa microbiome is involved in SS etiology. Regarding the gut microbiota, Moon J et al. showed that gut dysbiosis was partly correlated to dry eye severity [71]. Furthermore, SS showed significant gut dysbiosis compared to controls and environmental dry eye syndrome, while dry eye patients showed compositional changes of gut microbiome somewhere in between Sjögren’s syndrome and controls. Cano Ortiz et al. demonstrated that the SS patients had gut dysbiosis associated with increased serum levels of proinflammatory cytokines including IL-6, IL-12, IL-17, and TNF-alpha (systemic inflammation) and zonulin (intestinal permeability) that resulted in increased systemic microbial exposure [72].

## 5. Oral Microbiome on Systemic Lupus Erythematosus

Systemic lupus erythematosus (SLE) is a prototypic autoimmune disease affecting multiple organs, particularly in females of childbearing age. SLE is characterized by loss of tolerance to various self-antigens with autoantibody production to nuclear and cytoplasmic antigens and immune complex deposition, reflecting dysregulation of both the innate and adaptive immune systems [73]. Among clinical features, oral manifestations are common (in 5–40% of cases) and often are the first SLE manifestation [74]. A recent study suggested that the local oral microenvironment is involved in the development of oral SLE lesions and/or systemic lesions [70]. Among environmental factors implicated in SLE pathogenesis, the role of the microbiome has gained increasing interest in the last years. Gut microbiota has been shown to be associated with an imbalance in the proportions of Th17 and Treg cells [75] in SLE patients while the potential participation of oral microbiome remains elusive.

Moreover, intestinal dysbiosis has been demonstrated, consisting of an imbalance of specific gut flora in SLE patients compared with healthy controls (with a significantly lower *Firmicutes/Bacteroidetes* ratio and an associated over representation of oxidative phosphorylation and glycan utilization pathways) [76].

Furthermore, it has been recently shown that a mechanism through which the bacteria may drive autoimmunity in SLE is by the production of amyloid–DNA complexes, which can trigger an immune cascade involving TLR9 stimulation, IFNs type 1 transcription, and antinuclear antibody production [77].

Nonetheless, the frequent and early involvement of the oral mucosa in SLE suggests that the local oral microenvironment may participate in the development of oral SLE lesions and/or may contribute to systemic involvement through the generation of circulating autoantibodies against oral microbial products. In addition, the viral infections (i.e., EBV and CMV) that have been implicated in SLE pathogenesis may also arise in the oral cavity. Li et al. found that in SLE patients, the oral microbiota was inbalanced and diversity was reduced but no difference was found between new-onset and treated SLE patients. The abundances of *Lactobacillaceae*, *Veillonellaceae*, and *Moraxellaceae* were increased in patients with systemic lupus erythematosus, whereas those of families were decreased, such as *Corynebacteriaceae*, *Micrococcaceae*, *Sphingomonadaceae*, *Halomonadaceae*, *Xanthomonadaceae* [78].

Finally, the genus *Veillonella* was found to be increased in SLE patients with oral ulcers. A recent study showed that the genera *Haemophilus* and *Veillonella* were among the main members of the oral microbiota in healthy adults [79]. Additionally, *Haemophilus parainfluenzae* was abundant in patients with Behçet’s disease [80]. The expression of oral ulcers in these autoimmune diseases may be related to overabundance of the genera *Veillonella* and *Haemophilus* in the oral environment.

SLE is also associated with alterations of gut microbiota. Fecal microbiome from SLE mice can induce the production of anti-dsDNA antibodies in germ free mice and stimulate the inflammatory response, and alter the expression of SLE susceptibility genes in these mice. The *Firmicutes*-to-*Bacteroidetes* ratio has been demonstrated to be consistently reduced in SLE patients, regardless of ethnicity. The relative abundance of Lactobacillus differs from the animal model used (MRL/lpr mice or NZB/W F1 mice). This may indicate that interactions between gut microbes and the host, rather than the enrichment of some gut microbes, are especially relevant for the SLE development. *Enterococcus gallinarum* and *Lactobacillus reuteri*, both of which are possible gut pathobionts, become translocated into systemic tissue if the gut epithelial barrier is impaired. The microbes then interact with the host immune systems, activating the type I IFN pathway and inducing autoantibody production. In addition, molecular mimicry may critically link the gut microbiome to systemic lupus erythematosus. Gut commensals of SLE patients share protein epitopes with the Ro60 autoantigen. *Ruminococcus gnavus* strain cross-reacted with native DNA, triggering an anti-double-stranded DNA antibody response [81]. Expansion of *R. gnavus* in SLE patients is paralleled to an increase in disease activity and lupus nephritis. Such insights into the link between the gut microbiota and SLE enhance our understanding of SLE pathogenesis and will identify biomarkers predicting active disease.

## 6. Oral Microbiota as Promising Diagnostic Biomarkers for Rheumatology Diseases

With the improvement in NGS tools available to investigators, we have a greater understanding of the dysbiotic oral microbiome diversity and its effect on a number of systemic pathologies, indicating the potential of oral microbiota as a non-invasive diagnostic tool. In general, the oral cavity would be an ideal site for analyzing biomarkers because samples are comparatively easy to obtain. Indeed, saliva is a non-invasive collection method that does not cause discomfort and pain to patients. To analyze the oral microbiome, each oral micro-habitat has to be sampled with an adequate methodology. For oral mucosa, the use of nylon sterile microbrushes [20,82] and sterile brushes [83] have been reported. Sterile Gracey curettes are used for teeth hard tissues [84] or plaque sampling on endodontics paper cones [85], sterile toothpicks [86], sterile microbrushes, and floss [86]. For saliva sampling, non-stimulated saliva [87] has been used, but other studies rather used oral rinse (saliva after rinsing), to obtain a higher fraction of microbes possibly adhered to oral surfaces [88].

Previous studies reported the possibility of clinical use of oral bacteria in various kinds of tumors such as pancreatic cancer, lung cancer, esophageal [89], and oral cancers [90,91]. Regarding rheumatology diseases, [53] as previously reported, recent studies suggest that RA has a correlation with oral microbiome and may be affected by its dynamic variations. A new research study of 2018 may provide the impetus for a RA diagnostic testing via biomarkers identified in the oral microbiome [92]. The investigators compared the oral microbiota profiles of RA patients, osteoarthritis (OA) patients, and healthy subjects, which showed significant differences between RA patients, OA patients, and the healthy controls. Using the information on the structural changes in the oral microbiota, eight oral bacterial biomarkers were identified to differentiate RA from osteoarthritis, notably *Actinomyces*, *Neisseria*, *Neisseria subflava*, *Haemophilus parainfluenzae*, *Haemophilus*, *Veillonella dispar*, *Prevotella*, and *Veillonella*. This report provides proof of oral microbiota as an informative source for discovering non-invasive biomarkers for arthritis screening.

In a very recent study, saliva samples were collected from RA high-risk individuals, who were positive for ACPA and have no clinical arthritis, from RA patients and healthy controls [93]. In the “pre-clinical” stages, salivary microbial diversity was significantly reduced compared to RA patients and healthy controls. Individuals at high-risk for RA showed a reduction in the abundance of genus *Defluviitaleaceae*_UCG-011 and the species *Neisseria oralis*, but an expansion of *Prevotella*_6. Interestingly, the authors observed a characteristic compositional change of salivary microbes in individuals at high-risk for RA, suggesting that oral microbiota dysbiosis occurs in the “pre-clinical” stage of RA and are correlated with systemic autoimmune features. These findings support the hypothesis that microbiome changes occurring in mucosal sites such as the oral cavity might contribute to disease pathogenesis in the initial RA stages. For this reason, manipulating the microbes by traditional dietary modifications, probiotics, and antibiotics and by currently employed disease-modifying agents seems to modulate the disease process and its “progression” [94].

Future studies should explore the use of oral microbiota dysbiosis as a biomarker of disease and the manipulation of oral microbiota therapeutically to change RA disease progression.

## 7. Future Direction: Artificial Intelligence (AI) and Its Application in the Prediction of the Link between Oral Microbiome and Rheumatic Diseases

Despite current failures in analysis and research design, advances in high-throughput sequencing techniques have created a significant quantity of sequencing data, bringing fresh insights into the oral microbiota and rheumatic diseases relationship. The next step would be to use AI techniques like machine learning (ML) and deep learning (DL) to predict rheumatic disease states from the oral microbiota. Consistent collection of host information across research is required to employ AI-based techniques for predicting oral microbiome–rheumatic diseases connections.

ML is a subset of AI that relies on mathematical models to discern trends and patterns from data without the use of explicit instructions [95]. ML-based methods that integrate multiple omics (for instance genomics, metabolomics, imaging, microbiome, etc.) and clinical data enable the definition of robust and sensitive multidimensional biomarkers related to complex diseases. In microbiome research, ML algorithms find important features from the given feature set (phylogeny, relative abundance, or functional profiles from the microbiome data) for more accurate prediction of the phenotype of interest, e.g., disease state.

In rheumatology, ML can represent a step towards precision medicine, leading to the improvement of patient profiling and treatment personalization [96,97]. Notably, ML can be split into two parts: supervised and unsupervised learning [95]. Supervised learning leads algorithms to solve a pre-defined problem. To provide a forecast, the ML software must be trained per acceptably large amount of data that had been previously labeled by humans [95]. It is also possible to identify the most relevant features for classification. On the other hand, supervised methods include decision tree algorithm, logistic regression, as well as more complex algorithms like random forest and artificial neural networks.

In recent years, both supervised and unsupervised methods were incrementally used in microbiome research. A glimpse of how precision microbiome medicine can become a reality was provided by Zeevi et al. who used a decision-tree with a gradient boosting model integrating 800 people’s blood parameters, dietary habits, anthropometrics, physical activity, and microbiome profile to predict postprandial glycemic response [98]. Similarly, in the oncology setting, supervised ML methods on microbiota data were used to predict chemotherapy effectiveness and tolerability [99]. ML had also been used to analyze how bacteria contribute to drug metabolism [100]. In oral microbiome research, ML has proven useful to predict atherosclerotic cardiovascular disease (ACVD). In a study including 43 patients with ACVD and 86 age- and sex-matched non-ACVD individuals, a random forest algorithm based on 43 unique operational taxonomic units (OTUs) was able to predict ACVD with area under the curve (AUC) of 0.93 [101]

In rheumatology, the use of ML in microbiome research remains largely unexplored. In this regard, although promising, the results seem less consistent than in other areas as of few studies with a limited sample size have been carried out only on fecal samples. In particular, in a cohort of 39 patients with juvenile idiopathic arthritis, a random forest algorithm was used to discriminate patients from healthy controls (AUC 0.80) by integrating information of 12 genera including *Anaerostipes*, *Dialister*, *Lachnospira*, and *Roseburia* from fecal samples [102]. Size and quality of datasets are of utmost importance in ML. 

The future of oral microbiome-systemic link studies relies on the development of larger datasets, meaning more consistent validation and testing performance. This is to say, in deploying any ML approach to microbiome in rheumatology, researchers actually face issues consisting in the scarcity of data sharing of rheumatic patient datasets. In contrast with other research areas such as oncology and dermatology [103,104], there is a lack of dedicated open repositories integrating either labeled clinical features, histopathologic visual contents, and/or microbiome data from patients with rheumatic diseases [97]. In the absence of large annotated datasets, the potential of ML methods involving oral microbiome data in predicting therapy response remains largely unexploited. For being finally adopted in real-life clinical settings and making microbiome-led precision medicine comes true in rheumatology, data and code sharing must be strongly advocated. The use of proper study designs and the gathering of detailed host personal data in future oral microbiome analysis could help to improve AI-based oral healthcare research, which could have significant clinical benefits such as the ability to predict systemic disease condition from the oral microbiota.

## 8. Conclusions

Microbiome research in rheumatic diseases is expanding significantly, offering unique opportunities to better understand aspects of pathogenesis, the potential for patient stratification, and its application towards personalized therapeutic strategies. Oral microbiota seems to be a promising diagnostic and prognostic biomarker and a useful tool that may help to guide the disease comprehension. Today, new perspectives in microbiome biomarkers research are represented by artificial intelligence approaches such as machine learning (ML). In rheumatology, AI can represent a step towards precision medicine, leading to the improvement of patient profiling and treatment personalization (Table 1 and Table 2).

## Figures and Tables

**Figure 1 jpm-11-00625-f001:**
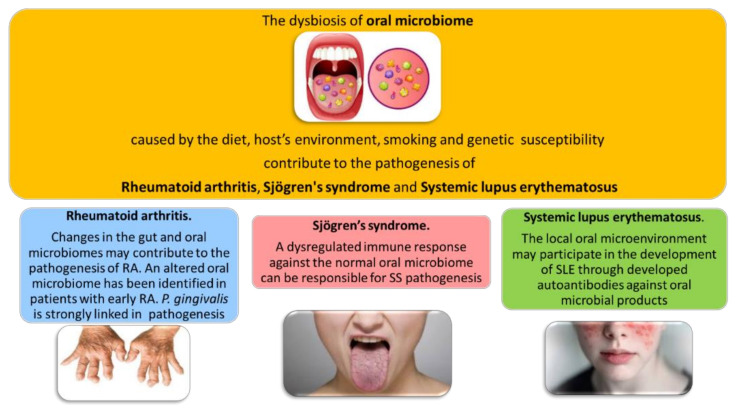
The dysbiosis of oral microbiome in rheumatic diseases.

**Figure 2 jpm-11-00625-f002:**
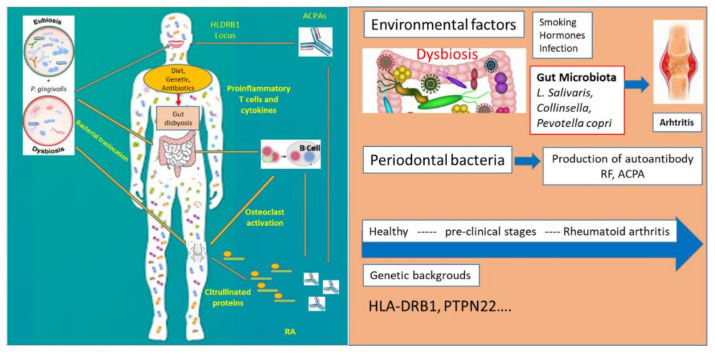
Genetic and environmental factors involved in the pathogenesis of rheumatoid arthritis.

**Figure 3 jpm-11-00625-f003:**
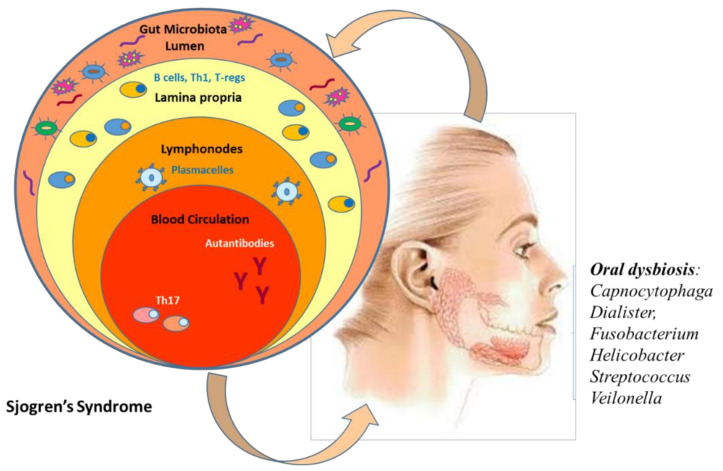
Oral dysbiosis in Sjogren’s syndrome.

**Table 1 jpm-11-00625-t001:** Literature data linking the oral microbiota with the pathogenesis of progression of rheumatic diseases.

Authors	Journal	Finding
Gomez-Banuelos, E.; et al. [35]	*J. Clin. Med.* **2019**	The subgingival microbiota differes significantly between RA and healthy individuals
Rosenstein, E.D.; et al. [37]	*Inflammation* **2004**	*P. gingivalis* expresses a bacterial protein arginine deiminase (PPAD) that can citrullinate free L- arginine and C-terminal arginine residues in cleaved peptides
Sato; et al. [39]	*Sci. Rep.* **2017**	*P. gingivalis* exacerbate arthritis by modulating the gut microbiota and increasing the proportion of Th17 (T helper 17) cells in mesenteric lymph nodes
Brusca, S.B.; et al. [45]	*Curr. Opin. Rheumatol.* **2014**	Several organisms, besides P. Gingivalis, cause periodontal disease (i.e., *Anaerglobus geminatus* and *Prevotella/Leptotrichia*) and are linked to the ACPA presence
Scher, J.U.; et al. [51]	*Arthritis Res. Ther.* **2013**	An alteration in the bacterial taxa of several mucosal sites (including oral, lung, and intestinal microbiomes) is required for the transition from a pre-clinical, autoimmune phase of RA into clinically classifiable disease
Ceccarelli, F.; et al. [55]	*Clin. Exp. Immunol.* **2018**	A significant association between the percentage of *P. Gingivalis* on the total tongue biofilm and RA disease activity (DAS28) was found
Szymula, A.; et al. [67]	*Clin. Immunol.* **2014**	A role for cross-reactivity of commensal oral and gut bacteria with SSA/Ro60 in the Sjogren Syndrome aetiopathogenesis
Horta-Baas, G.; et al. [86]	*J. Immunol. Res.* **2017**	Both gut and oral microbiota differ in early stages of RA from healthy controls, with a reduction of *Bifidobacterium* and *Bacteroides* and an increase in *Prevotella*
van der Meulen, T.A.; et al. [14]	*Rheumatology (Oxford)* **2018**	Buccal mucosa microbiome of primary Sjogren Syndrome (pSS) and non-SS sicca patients differ from healthy controls, with a higher *Firmicutes/Proteobacteria* ratio observed in both SS and non-SS sicca patients.
Greiling, T.M.; et al. [69]	*Sci. Transl. Med.* **2018**	*Corynebacterium amycolatum* has been shown to colonize the lacrimal duct, making *C. amycolatum* Ro60, a candidate ortholog for the development of anti-Ro60 antibodies in SS
Li, B.Z. et al. [78]	*Arch. Oral Biol.* **2020**	In SLE patients, the oral microbiota was imbalanced and diversity was reduced but no difference was found between new-onset and treated SLE patients

**Table 2 jpm-11-00625-t002:** Literature data in which AI tools is used to study the oral-gut microbiota axis of rheumatic diseases.

Authors	Disease	Number of Patients	Algorithm	Outcome
Qian, X.; et al. [102]	Juvenile Idiopathic Arthritis	39	Random Forest	Discrimination between patients and healthy controls

## Data Availability

Not applicable.

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
