# Peer review of "Exploring the Oral Microbiome in Rheumatic Diseases, State of Art and Future Prospective in Personalized Medicine with an AI Approach"

_jpm, 2021, doi:10.3390/jpm11070625_

Round 1

Reviewer 1 Report

The paper by Randone et al was an interesting and mostly well written review of the potential link between the oral and gut microbiomes and rheumatic diseases.   Some alternate analysis viewpoints need to be built upon (see the specific points below), although the literature coverage was good.  However, the machine learning section does not add to the review, not least because it did not focus sufficiently on the main review subject area.  The review should also be checked for text formatting as there are several sets of 1-2 sentence paragraphs.

Specific points

Line 156-157 ‘Another proof of the link between RA and oral inflammation is the evidence that the treatment of periodontal disease may improve RA symptoms’.

This needs to be referenced as it is a key point linking oral microbiome residents to RA.

Line 159-160 ‘Furthermore, microbiome of periodontally healthy individuals with and without RA has been evaluated. The subgingival microbiota differed significantly between RA and 160 healthy individuals.’

And section 6. Oral Microbiota as Promising Diagnostic Biomarkers for Rheumatology diseases

Oral microbiome species diversity profiles are linked to diet, ditto as are those of the gut microbiome, which has not been considered in connecting RD disease with microbiome residents. What would have been useful is to cover if changing the microbiome affects RD conditions, say though diet.  Also not covered is how medication used to treat RA disease might cause the microbiome differences observed.   The presence of a microbe or group of microbes does not mean they are the causative links to a disease, as it may be the conditions of the disease itself (diet, immune system changes, drug metabolism) which causes local species profile changes.

Line 405 Section 7, New Perspectives in the research of rheumatic diseases: microbiome biomarkers 405 through the Machine Learning approach

This technique analysis is out of place and does not substantially add to a review on linking microbiomes to rheumatic diseases. 

Author Response

The paper by Randone et al was an interesting and mostly well written review of the potential link between the oral and gut microbiomes and rheumatic diseases. Some alternate analysis viewpoints need to be built upon (see the specific points below), although the literature coverage was good.  However, the machine learning section does not add to the review, not least because it did not focus sufficiently on the main review subject area. The review should also be checked for text formatting as there are several sets of 1-2 sentence paragraphs.

Specific points

1 Line 156-157‘ Another proof of the link between RA and oral inflammation is the evidence that the treatment of periodontal disease may improve RA symptoms’.

This needs to be referenced as it is a key point linking oral microbiome residents to RA.

We thank the reviewer for the suggestion and we add the following references in the text:

BiaĹ‚owÄ…s K, Radwan-Oczko M, DuĹ›-Ilnicka I, Korman L, Ĺšwierkot J. Periodontal disease and influence of periodontal treatment on disease activity in patients with rheumatoid arthritis and spondyloarthritis. Rheumatol Int. 2020 Mar;40(3):455-463. doi: 10.1007/s00296-019-04460

2 Line 159-160‘Furthermore, microbiome of periodontally healthy individuals with and without RA has been evaluated. The subgingival microbiota differed significantly between RA and 160 healthy individuals.’

We thank the reviewer to have highlighted the incorrect sentences; we modified it and added reference:

“Furthermore, microbiome of periodontally healthy individuals with and without RA has been evaluated. The subgingival microbiota differed significantly between RA and healthy individuals”

Kaja Eriksson, Guozhong Fei, Anna Lundmark, Daniel Benchimol, Linkiat Lee, Yue O. O. Hu, Anna Kats, Saedis Saevarsdottir, Anca Irinel Catrina, Björn Klinge, Anders F. Andersson, Lars Klareskog, Karin Lundberg, Leif Jansson, and Tülay Yucel-Lindberg Periodontal Health and Oral Microbiota in Patients with Rheumatoid Arthritis J Clin Med. 2019 May; 8(5): 630. doi: 10.3390/jcm8050630 

And section 6. Oral Microbiota as Promising Diagnostic Biomarkers for Rheumatology diseases

Oral microbiome species diversity profiles are linked to diet, ditto as are those of the gut microbiome, which has not been considered in connecting RD disease with microbiome residents. What would have been useful is to cover if changing the microbiome affects RD conditions, say though diet. Also not covered is how medication used to treat RA disease might cause the microbiome differences observed. The presence of a microbe or group of microbes does not mean they are the causative links to a disease, as it may be the conditions of the disease itself (diet, immune system changes, drug metabolism) which causes local species profile changes.

We thank the reviewer for the interesting comment in fact the link between microbiome and medication and diseases is really complex and never unidirectional. We better specified the fact the microbiota may also predict response to treatment. The following sentences were added in the text:

“Furthermore, studies have shown that RA patients have altered gut and mouth microbiomes that are partly normalized after disease modifying antirheumatic drugs (DMARDs) and could predict response to treatment[i]. Zhang, X. et al.  shownthat patients that responded well to treatment were characterized by a greater number of virulence factors before treatment and also by the reduction in Holdemania filiformis and Bacteroides sp. after treatment. They also reported that the effect on the gut microbiome was moderate compared to the oral microbiome”

“For this reason , manipulating the microbes by traditional dietary modifications, probiotics, and antibiotics and by currently employed disease-modifying agents seems to modulate the disease process and its progression”

Line 405 Section 7, New Perspectives in the research of rheumatic diseases: microbiome biomarkers 405 through the Machine Learning approach

This technique analysis is out of place and does not substantially add to a review on linking microbiomes to rheumatic diseases.

We thank the reviewer for the right comment, we changed the name of the paragraph, the text has been rewritten and adapted to better suit the theme of the review, following the topic of “personalized medicine”

Reviewer 2 Report

Authors tried to review the relationship between oral and gut microbiome.

The viewpoint is significant and interesting, but there are several problems for the publication.

The title of this review is shown as “Exploring the oral-gut microbiome axis”. However, oral and gut microbiome are separately discussed and both relationship is unclear. Authors should focus on this relationship. If not, there is already reported review for the microbiome in rheumatic disease, such as Nat. Rev. Rheumatol. 17; 224-237, 2021. Reported review covers many articles referred in this article.

Totally authors mostly discussed the oral microbiome in rheumatic disease as shown like Figure 3.  If authors want to put the title like this submitted article, they should discuss the gut microbiome in rheumatic disease.

Author Response

Authors tried to review the relationship between oral and gut microbiome.

The viewpoint is significant and interesting, but there are several problems for the publication.

The title of this review is shown as “Exploring the oral-gut microbiome axis”. However, oral and gut microbiome are separately discussed and both relationship is unclear. Authors should focus on this relationship. If not, there is already reported review for the microbiome in rheumatic disease, such as Nat. Rev. Rheumatol. 17; 224-237, 2021. Reported review covers many articles referred in this article.

Totally, authors mostly discussed the oral microbiome in rheumatic disease as shown like Figure 3.  If authors want to put the title like this submitted article, they should discuss the gut microbiome in rheumatic disease.

We thank the reviewer for the observation, and we absolutely agree with him/her. In agreement with this comment, we focussed our review only in oral microbiome, all changes in the text are reported in yellow. Furthermore, we changed the title of the review.

Our review stands out from Nat. Rev. Rheumatol. 17; 224-237, 2021 because, after discussing the multiple links between oral microbiome and rheumatic diseases, we speculate a future application in personalized medicine.

Round 2

Reviewer 1 Report

I am satisfied the authors have made the required amendments.  However, a careful proof reading for grammar and sentence construction is still needed.

Reviewer 2 Report

The revised article was well-improved. No further revision was needed.